# Ileal Pouch–Anal Anastomosis and Pouchitis: The Role of the Microbiota in the Pathogenesis and Therapy

**DOI:** 10.3390/nu14132610

**Published:** 2022-06-24

**Authors:** Roberto Gabbiadini, Arianna Dal Buono, Carmen Correale, Antonino Spinelli, Alessandro Repici, Alessandro Armuzzi, Giulia Roda

**Affiliations:** 1IBD Center, Department of Gastroenterology, IRCCS Humanitas Research Hospital, Via Manzoni 56, 20089 Rozzano, Milan, Italy; roberto.gabbiadini@humanitas.it (R.G.); arianna.dalbuono@humanitas.it (A.D.B.); Carmen.Correale@humanitasresearch.it (C.C.); antonino.spinelli@hunimed.eu (A.S.); alessandro.repici@hunimed.eu (A.R.); alessandro.armuzzi@hunimed.eu (A.A.); 2Department of Biomedical Sciences, Humanitas University, Via Rita Levi Montalcini 4, 20072 Pieve Emanuele, Milan, Italy; 3Division of Colon and Rectal Surgery, IRCCS Humanitas Research Hospital, Via Manzoni 56, 20089 Rozzano, Milan, Italy

**Keywords:** inflammatory bowel diseases, ileal pouch–anal anastomosis, microbiota, dysbiosis, pouchitis

## Abstract

Inflammatory bowel diseases, Crohn’s disease and ulcerative colitis, are life-long disorders characterized by the chronic relapsing inflammation of the gastrointestinal tract with the intermittent need for escalation treatment and, eventually, even surgery. The total proctocolectomy with ileal pouch–anal anastomosis (IPAA) is the surgical intervention of choice in subjects affected by ulcerative colitis (UC). Although IPAA provides satisfactory functional outcomes, it can be susceptible to some complications, including pouchitis as the most common. Furthermore, 10–20% of the pouchitis may develop into chronic pouchitis. The etiology of pouchitis is mostly unclear. However, the efficacy of antibiotics in pouchitis suggests that the dysbiosis of the IPAA microbiota plays an important role in its pathogenesis. We aimed to review the role of the microbiota in the pathogenesis and as a target therapy in subjects who develop pouchitis after undergoing the surgical intervention of total proctocolectomy with IPAA reconstruction.

## 1. Introduction

In humans, a wide number of different microbial species are located in the bowel, which hosts several trillion microbial cells [1]. This ensemble of microbial species is generally called gut microbiota [2]. There is a profound interplay between the gut microbiota and the human biology [3]. Indeed, the gut microbiota is important for various physiological functions such as eliciting immune maturation [4], defending against pathogens colonization and overgrowth [5], influencing epithelial proliferation [6] and intestinal vascular density [7], modifying bile acids in the large bowel [8], promoting metabolic homeostasis [9] and hormone modulation [10], synthesizing vitamins [11] and neurotransmitters [12], supplying energy [1], and regulating bone metabolism [13]. The intestinal microbiota of a healthy subject is composed predominantly by *Bacteroidetes* and *Firmicutes*, with also other smaller sections comprised by *Actinobacteria*, *Proteobacteria*, *Verrucomicrobia*, methanogenic archaea, *Eucarya*, and various phages [1]. Modifications in the constitution and function of gut microbes lead to dysbiosis [1], and several diseases are associated with gut dysbiosis [1,14]. In particular, intestinal dysbiosis is an important feature of inflammatory bowel diseases (IBD) [14,15], playing a crucial role in the onset of the disease in predisposed subjects [16]. Indeed, some important alterations of intestinal microbiota have been identified in IBD, including an underrepresentation of *Firmicutes* (in particular *Faecalibacterium prausnitzii*) [17], *Bacteroidetes,* and *Lactobacillus* [16] with increased levels of *Proteobacteria* [18]. IBD are life-long disorders characterized by the chronic relapsing inflammation of the gastrointestinal tract [19,20] with the intermittent need for escalation treatment, eventually requiring surgical intervention [21]. In particular, although decreasing over time, subjects affected by UC still have a 5- and 10-year risk of colectomy of 7.0% and 9.6%, respectively [22]. Indications for colectomy comprise refractory acute severe UC, medically refractory disease, and colorectal cancer [19]. For these cases, restorative total proctocolectomy with ileal pouch–anal anastomosis (IPAA) is the surgical intervention of choice [19,23,24]. Although IPAA provides a good quality of life and satisfactory functional outcomes [25,26], it can be subject to some complications, including pouchitis as the most common [27]. Pouchitis is an active, non-specific, idiopathic inflammation of the IPAA mucosa [28]. Approximately 25% of subjects develop pouchitis a year after IPAA with an increasing trend that reaches up to 45% at 5 years [29]. Approximately 10–20% of the pouchitis may also progress to chronic pouchitis, leading to antibiotic dependency or refractoriness requiring immunosuppressive therapy [30]. Furthermore, pouchitis is a risk factor for hospitalization [31] and pouch failure [32], which can occur in 5–10% of cases [33,34,35]. The etiology of pouchitis is mostly unclear. However, the efficacy of antibiotics in pouchitis suggests that the IPAA-related dysbiosis of the microbiota could play an important role in its pathogenesis [36,37,38]. We aimed to perform a comprehensive review on the role of the microbiota in the pathogenesis and as a target therapy In subjects with ulcerative colitis who develop pouchitis after undergoing total proctocolectomy with IPAA reconstruction.

## 2. Methods

We performed a literature search in the PubMed database. The key words used were: “ileal pouch-anal anastomosis”, “pouchitis”, “microbiota”, and “dysbiosis”. Furthermore, references of original articles and relevant reviews were screened to find further publications.

## 3. IPAA Microbiota Evolution over Time

Although derived from the small intestinal tissue, the microbiota of the IPAA changes over time into a microbiota with a colonic profile [39,40,41,42]. These modifications can arise as early as two months after surgery and achieve a more stable composition as the years go by after the creation of the IPAA [40,42]. *Clostridium coccoides*, *Clostridium leptum*, *Bacteroides fragilis*, *Atopobium*, *E. coli*, *Klebsiella*, *Veillonella*, *Staphylococcus (coag-)*, and *Enterobacter* are the more counted bacterial species in functional IPAA [40,41,43]. In particular, it seems that the microbiota of healthy IPAAs try to recover to a composition comparable to that observed prior to surgery [43], and it has been hypothesized that the presence of *Veillonella*, *Lachnospiraceae, Ruminococcus gnavus,* and clostridial cluster IV (i.e., *Faecalibacterium prausnitzii*) might be a marker of regularity of the IPAA flora [37,43,44]. Furthermore, a comparison between the microbiota of IPAA in subjects with UC and subjects with familial adenomatous polyposis (FAP), which exhibit a low incidence of pouch inflammation, might help to understand the microbial families potentially implicated in the pathogenesis of pouchitis [37]. Indeed, a higher presence of sulfate-reducing bacteria (SRB) in UC-IPAA has been observed compared to FAP-IPAA [45,46]. SRB produce hydrogen-sulfide, which inhibits butyrate oxidation and prevents its utilization by the intestinal epithelial cells, potentially resulting in the damage of the mucosa of IPAA [37,46]. Other findings confirm the presence of differences between UC-IPAA and FAP-IPAA observing less bacterial diversity, an increased proportion of *Proteobacteria,* and decreased levels of *Bacteroidetes* and *Faecalibacterium prausnitzii* in the UC-IPAA group [47,48]. Table 1 shows the most common microorganisms and the main differences between healthy adults, IBD patients, UC-IPAA, and FAP-IPAA patients.

## 4. IPAA Microbiota and Pouchitis

It has been shown that patients with IPAA experiencing pouchitis exhibit a lower bacterial diversity [47,49,50,51] with an increased amount of aerobes and a decrease in anaerobes [36,37]. In particular, subjects with pouchitis display decreased levels of *Bacteroidetes* [50,52,53], *Lachnospiraceae* [44,49], *Ruminococcaceae* [49], *Streptococci* [47,54], and *Faecalibacterium* [49], while incremented levels of *Enterobacteriaceae* (including *E.Coli*) [47,49] *Fusobacterium* [54,55], and *Propionibacterium acnes* [56] have been observed. Furthermore, Lim et al. identified seventeen operational taxonomic units that were seen exclusively in subjects with inflamed pouches including *Desulfosporosinus*, *Leptospira*, *Microcystis*, *Methylobacter*, and *Pseudoalteromonas* [57], while Pawelka et al. observed that the presence of *Staphylococcus aureus* correlated with a higher degree of chronic inflammation [58]. *Ruminococcaceae* and *Lachnospiraceae* have an important role in preserving the host’s health due to their ability to hydrolyze starch and other sugars producing butyrate and other short-chain fatty acids (SCFAs), which are considered the principal nutrients of the colonic epithelial cells [59]. When pouchitis occurs, lower levels of SCFAs have been observed in contrast to uninflamed IPAA [37,60], and antibiotic therapy during pouchitis is associated with an increment of SCFAs [61]. Furthermore, *Lachnospiraceae* and *Ruminococcaceae* families are the only few bacteria with the ability to produce secondary bile acids (deoxycholic acid-DCA, lithocholic acid-LCA) through the 7α-hydroxylation of the primary bile acids (PBAs). Indeed, lower levels of *Ruminococcaceae* and secondary bile acids (SBAs) have been demonstrated in UC-IPAA patients in comparison to FAP-IPAA patients. These findings could partially explain the role of dysbiosis in pouchitis as it is known that SBAs can reduce both acute and chronic colitis in animal models and may be essential in preserving immune homeostasis in the IPAA [48]. 

Another difference that has been observed between UC-IPAA and FAP-IPAA patients is that the former group displays a higher presence of SRBs, which are known to reduce butyrate oxidation and intestinal nutrients, potentially damaging the IPAA mucosa. Indeed, SRBs have been linked to pouchitis [37,62]. The presence of SRBs may be due to the degradation of sulfomucin by commensal bacteria, resulting in the production of free sulfate and SRB colonization [37]. Additionally, in this case, treatment with antibiotics during pouchitis is associated with a reduction in hydrogen sulfide and SRBs [62]. These studies show that the microbiota undoubtedly plays a role in the inflammation of the IPAA. However, future longitudinal studies in the same subjects could reduce the impact of other factors on microbiota alteration (i.e., diet, drugs) [37,42]. Table 2 summarizes the studies that investigated the microbiota changes in UC-IPAA patients with pouchitis.

## 5. Microbiota as a Target for the Treatment of Pouchitis

### 5.1. Antibiotics

Antibiotics represent the mainstay for the treatment of pouchitis [63] as they can induce remission by 74% in chronic pouchitis [64]. Ciprofloxacin and metronidazole are the first-line recommended antibiotics, although the best modality of treatment is still unclear [65]. A randomized clinical trial showed that both antibiotics produced a reduction in the total Pouchitis Disease Activity Index (PDAI) score in patients with acute pouchitis; however, ciprofloxacin produced a greater reduction in both symptom score and endoscopic score with fewer adverse events compared to metronidazole (0% vs. 33%, respectively) [66]. Furthermore, it has been shown that a combination of the two antibiotics can be effective also in patients with refractory/recurrent pouchitis [67]. The mechanisms implicated in the pouch microbiota’s changes after antibiotic therapy that are responsible for their favorable effects are not clearly understood [38]. In the study of Gosselink et al., ciprofloxacin could eradicate pathogens that are significantly increased during pouchitis (*Clostridium perfringens*, hemolytic *Escherichia coli*) while not disrupting most of the anaerobic bacteria that contribute to the stability of the IPAA’s flora [36]. Subsequently, Dubinsky et al. observed that the effectiveness of antibiotic therapy in pouchitis may be ascribed to the establishment of an intestinal microbiota with non-pathogenic, antibiotic-resistant bacteria with low inflammatory potential. This newly established microbiota may prevent more aggressive inflammatory bacteria from colonizing the pouch [68]. However, within three months after therapy discontinuation, most subjects relapsed, requiring additional antibiotic treatment [68]. Indeed, 7–20% of subjects that experience a first episode of pouchitis will eventually develop chronic pouchitis [69]. Therefore, further interventions following treatment with antibiotics should be contemplated (i.e., probiotics, diet) to support beneficial bacteria and prevent subsequent colonization by pathogenic species [68]. 

### 5.2. Probiotics

Randomized placebo-controlled trials (RCTs) showed that a probiotic mixture of *Lactobacilli* (four strains), *Bifidobacteria* (three strains), and *Streptococcus thermophilus* was effective in maintaining remission in subjects that suffered previous pouchitis. The treatment was generally well tolerated without significant serious adverse events (only one patient stopped medication complaining of abdominal cramps, vomiting, and diarrhoea) [70,71]. The same probiotic mixture led to increased fecal levels of *lactobacilli*, *bifidobacteria,* and *Streptococcus salivarius* in comparison to patients treated with placebo (*p* < 0.001) [70]. Similar findings were observed also in the RCT of Mimura et al. [71]. One of the potential mechanisms of the beneficial effect of the probiotic mixture could be its ability to increase tissue levels of the anti-inflammatory interleukin (IL)-10 and to reduce levels of tumor necrosis factor (TNF)-α, interferon-γ, and IL-1α. IL-10 may increase the tolerance of the intestinal immune system to resident pouch bacteria. Nevertheless, other mechanisms are probably involved in the anti-inflammatory effects of probiotics [72]. Indeed, Persborn et al. demonstrated that maintenance treatment with Ecologic 825 (another probiotic mixture containing strains of *Lactobacilli* and *Bifidobacterium*) after induction therapy with antibiotics restored the mucosal barrier to *E.Coli* in subjects with pouchitis [73].

Interestingly, some probiotics have been shown to be effective also as a prophylaxis therapy after surgery, reducing the rate of the first episode of pouchitis [74,75]. Therefore, some Authors suggest prescribing prophylaxis treatment with probiotics in subjects with high-risk factors of pouchitis after surgery (i.e., primary sclerosing cholangitis and extraintestinal manifestations) [69].

In addition to probiotics, other treatments have been studied to rebalance the intestinal flora in patients with IPAA.

### 5.3. Fecal Microbiota Transplantation

Fecal microbiota transplantation (FMT) has been shown to be a successful treatment in other conditions of microbiota alteration, such as recurrent *Clostridioides difficile* infection [76,77]. As a result, there is increasing interest in the use of FMT to treat pouchitis. In the study of Kousgaard et al., FMT could increase the microbial diversity in subjects with chronic pouchitis and obtain clinical remission in 33% of the patients at 6 months of follow-up [78]. However, a recent systematic review observed that FMT seems ineffective in treating chronic pouchitis [79]. Indeed, two recent randomized controlled trials observed a low efficacy of FMT in chronic pouchitis. Interestingly, the majority of the relapses occurred during or shortly after the completion of FMT [80,81]. Overall, only a few studies have explored the role of FMT in chronic pouchitis so far, exhibiting some pitfalls such as the heterogeneity in study design and type of fecal transplant delivery [38]. Future specifically dedicated RCTs with large sample-sizes and standardized protocols (i.e., disease definitions, type of FMT delivery, dose, or duration) will help to ensure reproducible data and provide higher quality of evidence on the real efficacy of FMT in the treatment of chronic pouchitis [82].

### 5.4. Diet and Prebiotics

Finally, it is important to keep in mind that diet and prebiotics may also help in modifying the gut microbial composition [83,84]. Prebiotics are substrates that are selectively utilized by host microorganisms conferring health benefits [85]. Welters et al. showed that three weeks of dietary supplementation with 24g of inulin increased IPAA butyrate concentration and reduced the inflammation of the pouch mucosa with a significant decrease in both endoscopic and histologic PDAI scores [86]. Fruit consumption may also be protective against pouch inflammation. Ianco et al. observed that a decreased consumption in fruit and vegetables may be associated with pouchitis [87]. Accordingly, in the prospective study of Godny et al. the reduction in fruit consumption over time was associated with pouchitis recurrence and with reduced microbial diversity. Interestingly, the consumption of 1.5 or more servings/day of fruit was associated with a reduced risk of developing pouchitis in the following year [88]. Fruits are a source of dietary fibers that can be fermented in SCFA and can increase the levels of fiber-degrading bacteria (i.e., *Faecalibacterium*), balancing the mucus-degrading bacteria and supporting the intestinal barrier function [88]. Dietary intervention in chronic pouchitis has also been investigated by McLaughlin et al. In particular, treatment with 28 days of an exclusive elemental diet improved the median clinical PDAI and changed the microbiota with a trend towards improved levels of *Clostridium Coccoides* and *Eubacterium rectale,* both producers of butyrate. However, there was no reduction in endoscopic or histologic inflammation, and the study comprised a small sample size, so the authors did not recommend an elemental diet as a therapy for pouchitis but only for the temporary reduction of symptoms [89]. Based on this evidence, fruits and vegetables seem to have protective effects against pouchitis, and physicians should encourage pouch patients to include them in their daily diet. The abundant supply of micronutrients in fruits and vegetables may play a role by modulating the microbiota and exercising anti-inflammatory effects [90]. In the future, more interventional trials are needed to clarify the role and the impact of diet as a treatment and prophylaxis of pouchitis. Figure 1 and Table 3 summarizes the possible therapeutic interventions involving microbiota for the treatment of pouchitis.

## 6. Conclusions

The pathogenesis of pouchitis remains mostly unclear. However, evidence has shown that the microbiota plays a fundamental role and that dysbiosis with a reduced microbial diversity is associated with pouch inflammation. Nevertheless, a clear understanding of whether inflammation causes an alteration in IPAA microbiota or vice versa is still lacking. However, active interventions based on rebalancing the pouch microbiota are currently the key strategy for the treatment of pouchitis. Antibiotics represent the gold standard treatment for inducing remission. However, due to the high rate of relapse and the risk of developing chronic pouchitis after a first episode of acute pouchitis, prophylactic treatment with probiotics in order to maintain remission after antibiotics treatment appears to be a successful strategy. High-evidence studies are warranted to understand the role of FMT and diet intervention to maintain the remission of the disease.

## Figures and Tables

**Figure 1 nutrients-14-02610-f001:**
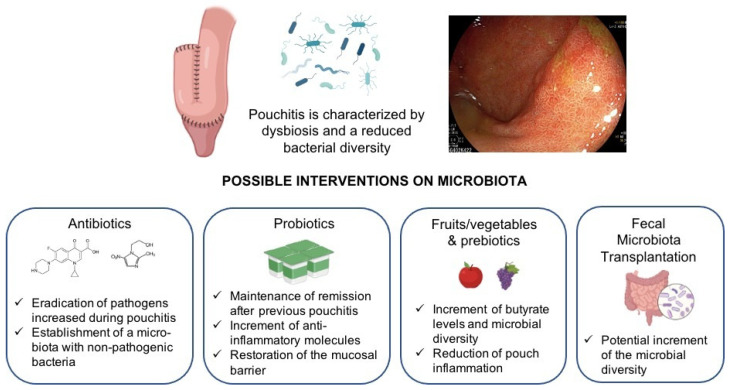
Interventions in the microbiota for the treatment of pouchitis. Antibiotic treatment (i.e., ciprofloxacin) may reduce the aerobic and pathogen bacteria (which are increased during pouchitis) while leaving undisturbed the larger part of the anaerobic flora, which is present in high numbers in subjects free of pouchitis [36]. Treatment with probiotics can increase fecal concentrations of anaerobes [70,75] and can decrease the levels of the *Escherichia* subgroup [75]. Pouch microbial diversity is also positively correlated with fruit consumption. In addition, the intake of fruit is positively correlated with the presence of Faecalibacterium and Lachnospira [88]. Finally, the treatment of pouchitis with fecal microbiota transplantation may increase the microbial diversity with a switch towards the donor’s microbiota [78].

**Table 1 nutrients-14-02610-t001:** Most common microorganisms and main differences between healthy adults, IBD patients, UC-IPAA, and FAP-IPAA patients.

Healthy Adults	IBD	UC-IPAA	FAP-IPAA
Bacteroidetes *	↓ Bacteroidetes	↓ Bacteroidetes	↑ Bacteroidetes
Firmicutes *	↓ Firmicutes	↓ Firmicutes	↑ Firmicutes
Actinobacteria *	↓ Lactobacillus	↑ Proteobacteria	↓ Proteobacteria
Proteobacteria *	↑ Proteobacteria	Presence of SRB	Absence of SRB
Verrucomicrobia	↑ Enterobacteriaceae		
Methanogenic archaea			
Eucaria (i.e., yeasts)			

* Representing about 90% of the bacterial phyla of the gut microbiota. IBD: inflammatory bowel disease; UC-IPAA: ulcerative colitis–ileal pouch–anal anastomosis; FAP-IPAA: familial adenomatous polyposis–ileal pouch–anal anastomosis; SRB: sulfate-reducing bacteria; ↓: reduced levels; ↑: increased levels.

**Table 2 nutrients-14-02610-t002:** Summary of the studies describing the alterations of the microbiota in UC-IPAA patients with pouchitis.

Authors	Sample Size	Bacterial Sequencing Platform	Results
Tannock [44]	34 UC-IPAA (17 with pouchitis), 14 FAP-IPAA	HTS of 16S rRNA genes (V1–V3 regions), FISH, qPCR	FAP-IPAA and UC-IPAA with normal pouch were more biodiverse than UC-IPAA with pouchitis (*p* < 0.0001) and displayed higher proportions of *Lachnospiraceae* (*p* < 0.05).
McLaughlin [47]	16 UC-IPAA (8 with pouchitis) and 8 FAP-IPAA (3 with pouchitis)	16S rRNA gene cloning and sequencing	Proteobacteria were increased (*p* = 0.019) while the Bacteroidetes were decreased (*p* = 0.001) in the total UC-IPAA compared with the total FAP-IPAA. *Faecalibacterium prausnitzii* was detected in the 75% of FAP-IPAA and only in 25% in UC-IPAA (*p* = 0.029). The median SDI for all UC-IPAA was 2.61 compared to 3.2 for all FAP -IPAA (*p* = 0.004)
Sinha [48]	17 UC-IPAA (4 with pouchitis), 7 FAP-IPA	LC-MS	UC-IPAA have reduced levels of lithocholic acid and deoxycholic acid (normally the most abundant gut SBAs) and *Ruminococcaceae* (one of few taxa known to include SBA-producing bacteria) compared to FAP-IPAA. SBA supplementation ameliorates inflammation in animal models of colitis.
Reshef [49]	131 UC-IPAA (83 with pouchitis and 10 with unstable pouch behavior), 9 FAP-IPAA	16S rRNA gene amplicon pyrosequencing	SDI was higher in normal UC-IPAA compared with UC-IPAA with pouchitis. *Faecalibacterium* was reduced in pouchitis compared to normal pouch (P 0.021). *Lachnospiraceae* and *Ruminococcaceae* were significantly decreased in pouchitis compared to normal UC-IPAA and FAP-IPAA.
Tyler [50]	53 UC-IPAA (15 with pouchitis and 19 with CDL), 18 FAP-IPAA	Pyrosequencing of the 16S rRNA V1-V3 hypervariable region	UC-IPAA had decreased microbial diversity compared to FAP-IPAA. *Bacteroidetes* were detected less frequently in UC-IPAA with pouchitis and CDL (*p* < 0.0001) compared to FAP-IPAA and UC-IPAA without pouchitis. *Proteobacteria* were detected more frequently in the inflammatory groups (pouchitis and CDL) (*p* = 0.01)
Li [51]	19 UC-IPAA (8 with pouchitis), 16 healthy controls, 41 UC.	Amplification of the V3 region of the 16S rRNA gene by the PCR technique	Healthy controls displayed a higher microbial biodiversity compared to UC-IPAA (*p* < 0.001). UC-IPAA with pouchitis showed fewer *Eubacterium rectale* (a butyrate-producing bacteria) and more *Clostridium perfringens* compared to UC-IPAA with normal pouch and healthy controls.
Zella [52]	12 UC-IPAA (9 with pouchitis), 7 FAP-IPAA	16S rDNA-based terminal restriction fragment length polymorphism	UC-IPAA with pouchitis exhibited less *Lactobacillus* and *Streptococcus* compared to FAP-IPAA. On the other hand, Bacteroidetes were higher in the FAP-IPAA compared to UC-IPAA with pouchitis (*p* < 0.001).
Iwaya [53]	22 UC-IPAA (9 with pouchitis)	Culture	*Bacteroidaceae*, *Bifidobacterium*, and *Lactobacilli* were significantly lower in UC-IPAA with pouchitis compared to UC-IPAA without pouchitis (*p* < 0.01, *p* < 0.001, and *p* < 0.05 respectively).
Komanduri [54]	20 UC-IPAA (5 with pouchitis), 13 healthy controls	16S rRNA–based LH-PCR	*Streptococci* were associated with UC-IPAA without pouchitis and were lower in pouchitis. Conversely, members of the *Fusobacterium* group were associated with pouchitis.
Petersen [55]	20 UC-IPAA (10 with pouchitis), 30 healthy controls, 140 IBD	16S rDNA MiSeq sequencing (V3–V4 region)	Higher levels of *Fusobacteria* were found in UC-IPAA with a fecal calprotectin level >500 (*p* = 0.02). UC-IPAA had a lower SDI compared to healthy controls and IBD. The SDI was not significantly different between UC-IPAA with active and inactive inflammation (*p* = 0.74).
Palmieri [56]	34 UC-IPAA (13 with pouchitis), 19 healthy controls	16S rDNA MiSeq sequencing (V3–V4 region)	UC-IPAA (with or without pouchitis) had significantly lower bacterial diversity compared to healthy controls. *Faecalibacterium prausnitzii* was more abundant in healthy controls than in the total UC-IPAA. *Propionibacterium acnes* was significantly associated with pouchitis.
Lim [57]	20 UC-IPAA (5 with pouchitis)	T-RFLP of 16S rDNA	Seventeen operational taxonomic units (OTU) were found exclusively in pouchitis; 11 of the 17 OTUs were entirely novel. The other six OTUs were identified as *Pseudoalteromonas, Desulfosporosinus, Methylobacter, Leptospira,* uncultered *proteobacterium*, and *Microcystis*.
Pawełka [58]	47 UC-IPAA (11 with pouchitis)	Culture	The presence of *Staphylococcus aureus* significantly correlated with a higher degree of chronic inflammation.
De Preter [60]	22 UC-IPAA (15 with pouchitis, 5 excluded pouch), 17 healthy controls	Butyrate oxidation: Biopsies incubation with 1 mM ^14^C-labeled Na-butyrate and measuring the released ^14^CO_2_.	Butyrate oxidation in UC-IPAA with mild or active pouchitis was decreased compared with that in normal ileum of healthy controls (*p* = 0.001) and in excluded/normal UC-IPAA (*p* = 0.005).
Sagar [61]	32 UC-IPAA (10 with pouchitis)	Measurement of stool concentrations of SCFA by gas–liquid chromatography	Stool concentrations of SCFA were lower in UC-IPAA with pouchitis compared to healthy UC-IPAA (*p* < 0.01). Resolution of pouchitis after antibiotic treatment (6 week course of metronidazole) was associated with an increment of SCFA stool concentration (*p* < 0.01).
Ohge [62]	45 UC-IPAA (19 with pouchitis), 5 FAP-IPAA	Measurement of hydrogen sulfide by gas chromatography. Serial tenfold dilutions of fecal homogenates for enumerating sulfate-reducing bacteria	FAP-IPAA produced significantly less hydrogen sulfide than nonantibiotic-treated UC-IPAA. Sulfate-reducing bacteria were significantly higher in UC-IPAA with pouchitis compared to UC-IPAA, which never experienced pouchitis. Sulfate-reducing bacterial count dropped with antibiotic treatment.

FAP-IPAA: familial adenomatous polyposis–ileal pouch–anal anastomosis; CDL: Crohn’s disease like; FISH: fluorescence in situ hybridization; HTS: High-throughput sequencing; IBD: inflammatory bowel diseases; LC-MS: liquid chromatography–mass spectrometry; LH-PCR: length heterogeneity polymerase chain reaction; qPCR: quantitative polymerase chain reaction; SBA: secondary bile acids; SCFA: short chain fatty acids; SDI: Shannon Diversity Index; rDNA: ribosomal DNA; rRNA: ribosomal RNA; T-RFLP: terminal restriction fragment length polymorphism; UC: ulcerative colitis; UC-IPAA: ulcerative colitis–ileal pouch–anal anastomosis.

**Table 3 nutrients-14-02610-t003:** Studies evaluating the interventions on the microbiota as a therapeutic target of pouchitis.

Authors	Study Design	Intervention	Results
Gosselink [36]	Observational	Ciprofloxacin (daily, 2 × 500 mg) or Metronidazole (daily, 3 × 500 mg) for two weeks	In subjects with pouchitis, metronidazole eradicated the anaerobic flora, including C. perfringens but not E. coli. Ciprofloxacin eradicated C. perfringens and E. coli, while the large part of the anaerobic flora was not changed. Ciprofloxacin produced a larger reduction in the PDAI score compared to metronidazole (*p* = 0.04).
Shen [66]	RCT	Ciprofloxacin (1000 mg/d) vs. Metronidazole (20 mg/kg/d) for two weeks	Both antibiotics significantly reduced total PDAI score in subjects with acute pouchitis. Ciprofloxacin produced a greater total PDAI reduction than metronidazole (6.9 vs. 3.8, *p* = 0.002). Ciprofloxacin group did not experience AEs, while the metronidazole group experienced AEs in 33% of cases.
Mimura [67]	Observational	Ciprofloxacin (500 mg b.i.d) plus metronidazole (400 or 500 mg b.i.d) for 28 days	Remission was obtained in 82% of cases of refractory or recurrent pouchitis. Median PDAI score was reduced from 12 to 3 (*p* < 0.0001).
Dubinsky [68]	Observational	Ciprofloxacin (500 mg b.i.d) plus Metronidazole (500 mg b.i.d) (two weeks courses or more in case of relapse)	In total, 79% of the antibiotic-treated subjects achieved clinical response. Antibiotics established an antibiotic-resistant microbiome with low inflammatory characteristics, which may confer resistance against colonization by bacteria that stimulate inflammation.
Gionchetti [70]	RCT	Probiotic mixture (6 g/day) of lactobacilli (4 strains), bifidobacterial (3 strains), and Streptococcus thermophilus vs. placebo for 9 months	All subjects with chronic pouchitis were in clinical and endoscopic remission at the start of the study. During the follow-up, 15% had relapses in the probiotic group compared with 100% in the placebo group (*p* < 0.001). Fecal concentration of lactobacilli, bifidobacteria, and S. thermophilus increased from baseline levels only in the probiotic group (*p* < 0.01).
Mimura [71]	RCT	Probiotic mixture (6 g/day) of lactobacilli (4 strains), bifidobacterial (3 strains), and Streptococcus thermophilus vs. placebo for 12 months or until relapse	Patients with refractory or recurrent pouchitis that achieved remission with a four week course of ciprofloxacin plus metronidazole were included at the start of the study. The cumulative maintained remission rate over the 12 months period was 85% in the probiotic group and 6% in the placebo group (*p* < 0.0001).
Gionchetti [74]	RCT	Probiotic mixture (3 g/day) of lactobacilli (4 strains), bifidobacterial (3 strains), and Streptococcus thermophilus vs. placebo for one year	The study included subjects with IPAA and started within 1 week after ileostomy closure. During the follow-up, 10% of patients in the probiotic group experienced an episode of acute pouchitis compared with the 40% of patients treated with placebo (*p* < 0.05).
Yasueda [75]	RCT	Probiotic (Clostridium butyricum MIYAIRI) (daily, 60 mg × 3) vs. placebo for 24 months.	The subjects included had not developed previous pouchitis after surgery; 11% in the probiotic group and 50% in the placebo group developed pouchitis (*p* = 0.07).
Kousgaard [78]	Observational	Fecal microbiota transplantation by enemas (20 g) for 14 consecutive days	Patients with chronic pouchitis were included, and 44% of patients were in clinical remission at 30-day of follow-up; 33% were in remission until 6 months of follow-up. FMT increased microbial diversity in 67% of subjects.
Herfarth [80]	RCT	Fecal microbiota transplantation by both enemas (24 g) and oral capsules (4.2 g) vs. placebo	Patients with antibiotic-dependent pouchitis were enrolled. The study was terminated prematurely due to a small clinical remission rate (6 subjects included). Only 1 patient (17%) clinically responded to FMT and remained off antibiotics for the study period of 16 weeks.
Karjalainen [81]	RCT	Fecal microbiota transplantation by endoscopy (30 g) and via transanal catheter (30 g) vs. placebo	Patients with chronic pouchitis were included and were followed up for 52 weeks; 34.6% patients in the FMT group and 30.8% in the placebo group relapsed during the follow-up (*p* = 0.183)
Welters [86]	RCT	24 g of inulin daily for three weeks vs. placebo	Butyrate concentrations were significantly higher in the intervention group compared to placebo. The endoscopic and histologic scores were lower in the inulin group compared with placebo.
Ianco [87]	Observational	Diet	Subjects with normal pouch consumed more fruits than subjects with pouchitis (3.6 s/d vs 1.8 s/d, *p* = 0.015) and tended to consume more vegetables (4.5 s/d vs 3.3 s/d, *p* = 0.06).
Godny [88]	Observational	Diet	Subjects in the lower tertile of fruit consumption (<1.45 s/d) had a higher rate of pouchitis within 1 year of follow-up, compared with patients in the upper two tertiles (30.8% vs 3.8%, *p* = 0.03). Fruit consumption was associated with an improved microbial diversity (*p* = 0.003).
McLaughlin [89]	Observational	Elemental diet (E028, SHS, UK) for 28 days	Elemental diet reduced the PDAI symptom score (from 4 to 1, *p* = 0.039). There was no reduction in endoscopic or histological signs of inflammation.

FMT: fecal microbiota transplantation; PDAI: pouchitis disease activity index; AE: adverse event; RCT: randomized clinical trial; s/d: servings/day.

## Data Availability

Not applicable.

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
