# Peer review of "Ileal Pouch–Anal Anastomosis and Pouchitis: The Role of the Microbiota in the Pathogenesis and Therapy"

_nutrients, 2022, doi:10.3390/nu14132610_

Round 1
Reviewer 1 Report
The problem reviewed in the paper is significant and deserves attention. However, in my opinion, the paper should be reconstructed beginning from the title and abstract to the conclusions, which start with the obvious statement about the unclear pathogenesis of pouchitis. The subsections regarding the interventions should be provided (diet, FMT etc.) . The methods of the paper search are missing.
Author Response
Reviewer 1:
The problem reviewed in the paper is significant and deserves attention. However, in my opinion, the paper should be reconstructed beginning from the title and abstract to the conclusions, which start with the obvious statement about the unclear pathogenesis of pouchitis. The subsections regarding the interventions should be provided (diet, FMT etc.) . The methods of the paper search are missing.
-Thank you for the advises for improving our work. Pouchitis is characterized by an unclear pathogenesis with dysbiosis playing an important role as we stated in the introduction of our work. The review is focused on the role of the microbiota and dysbiosis in the etiology of pouchitis, so we felt that it was necessary to include a brief general explanation about the microbiota and about the IBD to better understand the issue discussed in our review.
-We added the subsections regarding the interventions
-We added the methods of the paper search .
Reviewer 2 Report
In this article the authors comprehensively reviewed literature studying the role of the microbiota in the pathogenesis and therapy of IPAA and pouchitis. They provide a great overview of all current studies
This article needs moderate changes in the English language
The article would benefit from separate headlines for antibiotics and probiotics in paragraph 4
Line 156. It is not clear here whether this is the result of the probiotic mixture or the antibiotic therapy. And was it a placebo for the probiotic mixture?
What are the downsides of chronic use of probiotics
Lines 172-185. The flow of this paragraph could be improved. Please also state how long patients would benefit from FMT.
Lines 186-210. There is no attention paid to the implications of profylaxis by nutrition. Fruits are easy to implement. What is the duration effect of all interventions. Fruit can be easily eaten every day.
Line 212 Figures are of low quality and not well outlined
In the conclusion focus is more on medication (antibiotics and probiotics) whereas food can play an important role to prevent chronic pouchitis. Authors should discuss more the possibilitis of profylaxis in order to avoid chronic pouchitis.
Author Response
Reviewer 2:
In this article the authors comprehensively reviewed literature studying the role of the microbiota in the pathogenesis and therapy of IPAA and pouchitis. They provide a great overview of all current studies
This article needs moderate changes in the English language
Thank you for the advise for improving our work. We revised the paper with a native English speaker
The article would benefit from separate headlines for antibiotics and probiotics in paragraph 4
Thank you for the suggestion. We added separate headlines.
Line 156. It is not clear here whether this is the result of the probiotic mixture or the antibiotic therapy. And was it a placebo for the probiotic mixture?
Thank you for the suggestion. It is a results for the probiotic mixture. We made the corrections to better understand the line.
What are the downsides of chronic use of probiotics
Thank you for the suggestion. The use of probiotics is generally safe. We added the data in the manuscript.
Lines 172-185. The flow of this paragraph could be improved. Please also state how long patients would benefit from FMT.
Thank you for the suggestions. We added the data concerning the duration of benefit (or lack of benefit) of the FMT to improve this paragraph.
Lines 186-210. There is no attention paid to the implications of profylaxis by nutrition. Fruits are easy to implement. What is the duration effect of all interventions. Fruit can be easily eaten every day.
Thank you for the suggestions for improving the manuscript. Dietary intervention in pouchitis is an interesting topic because it might help prevent the development of intestinal inflammation. We added in the manuscript how the consumption of >=1.5 fruit servings/day was associated with a reduced risk of developing pouchitis within a year and advised physicians to encourage patients to eat fruit.
Line 212 Figures are of low quality and not well outlined
Thank you for the suggestion. We improved the alignment of the figure. We used a 300 DPI JPEG format as requested by the journal.
In the conclusion focus is more on medication (antibiotics and probiotics) whereas food can play an important role to prevent chronic pouchitis. Authors should discuss more the possibilitis of profylaxis in order to avoid chronic pouchitis.
Thank you for all the suggestions. To the best of our knowledge, probiotics have the higher evidences on the efficacy of prophylaxis of pouchitis. Therefore, we suggested the use of probiotics for prophylaxis of pouchitis, encouraging also the dietary interventions (fruit) that will need more well-designed studies to better understand their role in maintaining the remission of the disease.
Reviewer 3 Report
In this review article, Gabbiadini et al. summarized the current understanding of the potential role of the gut microbiota in the pathogenesis of IPAA-related pouchitis.
Major comments:
1. Line 67: "IPAA microbiota evolution over time", this part seems not to talk about the evolution of microbes at all.
2. Table 1: This table is not informative enough. Authors should prepare a table to summarize the key references to support those IPAA-related changes in the human microbiome, including sample size, ethnicity, sequencing platform, key findings, etc.
3. Fig.1: Pouchitis is characterized by a reduced bacterial diversity, did those interventions (antibiotics, prebiotics, probiotics, and FMT) help regarding alpha diversity?
4. Table2: the dose of treatment should be indicated.
Minor comments:
1. "dysbiosis of the IPAA microbiota", should be " IPAA related dysbiosis of microbiota"?
2. Line 174: "Clostridium difficile infection" should be " Clostridioides difficile infection"
Author Response
Reviewer 3:
In this review article, Gabbiadini et al. summarized the current understanding of the potential role of the gut microbiota in the pathogenesis of IPAA-related pouchitis.
Major comments:
- Line 67: "IPAA microbiota evolution over time", this part seems not to talk about the evolution of microbes at all.
Thank you for the suggestion. In this paragraph we described the microbiota of healthy pouches. We reported the changes of the IPAA flora over time and the main differences with the microbiota of the IPAA for familial adenomatous polyposis to better understand the normal findings of the pouch microbiota in ulcerative colitis (UC). We also reported studies which identified bacteria that might be a marker of regularity of the UC- IPAA flora.
- Table 1: This table is not informative enough. Authors should prepare a table to summarize the key references to support those IPAA-related changes in the human microbiome, including sample size, ethnicity, sequencing platform, key findings, etc.
Thank you for the suggestion. We added a table (Table 2) that summarizes the studies which investigated the microbiota changes in UC-IPAA patients with pouchitis
- Fig.1: Pouchitis is characterized by a reduced bacterial diversity, did those interventions (antibiotics, prebiotics, probiotics, and FMT) help regarding alpha diversity?
Thank you for the suggestions. These type of treatments can be effective in pouchitis due to their effect on pouch microbial composition. We added these informations in the figure caption
- Table2: the dose of treatment should be indicated.
Thank you for the suggestion. We added the dose of the treatments
Minor comments:
- "dysbiosis of the IPAA microbiota", should be " IPAA related dysbiosis of microbiota"?
Thank you for the suggestion. We made the correction
- Line 174: "Clostridium difficile infection" should be " Clostridioides difficileinfection"
Thank you for the suggestion. We made the correction
Round 2
Reviewer 3 Report
The manuscript has significantly improved and the authors have satisfyingly answered all my comments.